# Calving Management: A Questionnaire Survey of Veterinary Subject Matter Experts and Non-Experts

**DOI:** 10.3390/ani11113129

**Published:** 2021-11-01

**Authors:** Anna Lisa Voß, Wolfgang Heuwieser, John F. Mee, Carola Fischer-Tenhagen

**Affiliations:** 1Clinic for Animal Reproduction, Faculty of Veterinary Medicine, Free University of Berlin, Koenigsweg 65, 14163 Berlin, Germany; anna.voss@fu-berlin.de (A.L.V.); w.heuwieser@fu-berlin.de (W.H.); 2Animal Bioscience Research Department, Moorepark Research Centre, P61 P302 Fermoy, County Cork, Ireland; John.Mee@teagasc.ie; 3Center for Protection of Experimental Animals, German Federal Institute for Risk Assessment, 12277 Berlin, Germany

**Keywords:** survey, veterinarians, dairy, calving management, calving prediction, signs of imminent parturition, maternity pen

## Abstract

**Simple Summary:**

We designed a questionnaire and asked two groups of veterinarians: (1) subject matter experts, who had published on calving management and (2) veterinary practitioners) for their opinion about aspects of calving management. Participants recommended to differentiate between the two stages of parturition and emphasized signs of imminent parturition, such as “restlessness” and “visibility of fetal parts”. There was no consensus on the right time to move the cow to the maternity pen. Almost half of the respondents recommended a 6-h observation interval for prepartum cows in the maternity pen. The two veterinary groups differed little in their knowledge of calving management.

**Abstract:**

Accurate detection of the onset of parturition is a key factor in the prevention of dystocia. In order to establish current best practice recommendations for calving management, we asked subject matter experts (SME) who had published on calving management (by online survey, *n* = 80) and non-SMEs, veterinary practitioners (by workshop survey, *n* = 24) for their opinions. For this, we designed a questionnaire on the significance of signs of imminent parturition (SIP), the frequency of calving observation, and influencing factors for the timing of cow movement to a maternity pen. The response rate was 67.5% in the online survey and 100% in the workshop survey. The majority (89.7%) of all respondents agreed that it is beneficial for successful calving management to differentiate between stage I and II of parturition. Of 12 signs of imminent parturition (for stage I and II), “restlessness” and “visibility of fetal parts in vulva” were cited by 56.5% and 73.3% of SME and non-SME respondents, respectively. There was no consensus on the right time to move the cow to the maternity pen; recommendations varied from one to over 21 days. Almost half of the respondents (45.7%) recommended a 6-h observation interval for prepartum cows in the maternity pen. This study identified a strong consensus on the SIP and how and when to observe cows prior to parturition. SMEs and non-SMEs provided broadly similar recommendations, while the SMEs and the non-SMEs differed significantly in the number of publications on calving they authored, they differed little in their knowledge of calving management.

## 1. Introduction

Between 2 and 10% of all calves are born dead or die in the next 48 h after birth [1]. To prevent stillbirth and consequences of dystocia and related diseases, experienced personnel is required to detect the onset of parturition [2,3]. Management of periparturient cows is a skill learned through education and experience. However, diverse recommendations are made both in the scientific literature and by veterinary practitioners, for example, for the time of moving cows to maternity pens or time of intervention. In addition, due to the wide variation between cows regarding onset and progression of external signs of parturition [4] even experienced personnel do not detect the onset of all calvings [5]. Though various monitoring devices have been developed for calving detection, visual observation of cow behavior is the most commonly adopted approach [6].

Maternity pens (i.e., separate dedicated areas where calving takes place) not only provide a lower risk of spreading infections [7], but they also minimize the stress level of the cow during parturition [8]. As cows are moved to the maternity pen based on the expected calving date or behavioral or physiological signs, time spent in the maternity pen can vary [9]. According to some authors, an early pen move allows cows to adapt to the new environment, new diet, and in the case of a group calving barn, to the social structure; as these are all stressors that can negatively influence calving performance, especially in heifers [10,11]. In contrast, Gygax et al., 2015 [8] could not find a positive influence of such prepartum exposure to the new environment on calving performance. Moving cows within one or two days pre parturition aligns with cows’ natural isolation-seeking behavior, and on the other hand, will not affect the cleanliness and management of the calving environment too much [12,13]. Other authors found that an early pen move (≥3 d before parturition) is associated with higher incidences of ketosis and displaced abomasum [14]. Moreover, early movement has been associated with dystocia and stillbirth [15], which in turn increases the likelihood of trauma to the cow (i.e., paresis), uterine disease, and decreased milk yield [16,17]. Conversely, if cows are moved too late, during the late first stage of parturition, the second stage may be prolonged [18] which may lead to complications during calving and a 2.5-fold increased risk of stillbirths [19,20]. Therefore, some authors recommend moving cows during stage II of parturition [19,21,22]; “just-in-time” calving.

Specific recommendations regarding the prediction of calving time and consequent movement to the maternity pen are rare. Relaxation of the pelvic ligaments [23] or the concentration of inorganic phosphorus in mammary secretion [24] were recommended for deciding when to move a cow to the maternity pen [24,25]. In practice, clear landmarks such as the amniotic sac or feet of the calf visible outside the vulva are widely used by veterinary practitioners for advice on the just-in-time movement of cows. This apparent gap between what is published by experts in the literature and what veterinary practitioners actually recommend for calving management has not been explored heretofore.

This raises the generic issue of how subject matter experts (SMEs), e.g., academics, might differ from non-SME, e.g., veterinary practitioners, in their recommendations. An adjacent study of causes of perinatal calf mortality revealed a surprising consensus between SME and non-SME [26]. Any potential knowledge discordance between SMEs and non-SMEs is important, as farmers rate veterinarian practitioners as a very important source of information [27,28,29].

Given this paucity and diversity of opinion in the literature and the knowledge gap of what veterinary practitioners recommend, the objective of this survey was to elicit current veterinary opinions on calving management in order to provide farmers with best practice recommendations for managing the prepartum cow. Furthermore, we wanted to find out, if there is a difference in opinion between SMEs and non-SMEs.

## 2. Materials and Methods

### 2.1. The Questionnaire

A comprehensive questionnaire in English was developed by a team of veterinary researchers focusing on calving management. The questionnaire was assessed for readability, clarity, and logical structure by 12 research associates before administration. Questions (*n* = 18) were either open (*n* = 6), semi-open (*n* = 4), multiple-choice (*n* = 1), or closed (*n* = 7). Four questions covered general information on the participants and 14 questions comprised a selection of signs of imminent parturition (SIP), definitions of stages of calving, and information on influencing factors during calving, technical devices, and management of dystocia. The original questionnaire is provided in the Supplementary Online Appendix A as Figure A1.

### 2.2. The Respondents

Two groups of veterinary respondents were surveyed; corresponding authors of peer-review papers on calving management (subject matter experts; SMEs) and veterinary practitioners (non-SMEs). To identify SMEs, two comprehensive literature searches were conducted, and the corresponding authors of publications were contacted. Three different search strategies were applied.

First, the Pubmed database (https://www.ncbi.nlm.nih.gov/pubmed/) was accessed (on 28 June 2018), using the keyword combination “Calving Management OR Calving Prediction” for relevant publications (*n* = 1283). Results were filtered by applying the following exclusion criteria (Figure 1). Publications had to be written in English (*n* = 1252), published between 1998 and 2018 (*n* = 1045) with the subject: veterinary science (*n* = 1031) and available as full text (*n* = 973). Publications with a title that was off-topic (i.e., a title that was unrelated to the content of calving management or calving prediction) (*n* = 801), published in a book (*n* = 1) or a journal with a 5-year citation index (IF) of ≤1.0 (*n* = 1), and dealt with beef cattle (*n* = 64) or species other than cattle (*n* = 24) were excluded. The 82 remaining publications were examined in more detail and selected as reference publications based on the study objective (*n* = 30). Corresponding authors that were named twice or more (*n* = 8) were filtered out. Given the limited number of corresponding authors (*n* = 22) identified in the Pubmed search, another search engine (https://scholar.google.com/) was accessed on 15 July 2018, and publications before 1998, with an IF ≤ 1, and publications dealing with beef cattle were included, resulting in an additional 39 corresponding authors. The systematic literature research via Pubmed and Google Scholar resulted in 61 corresponding authors that were included in the survey.

Secondly, a supplementary search of the references (*n* = 319) in a convenience sample of 5 of the 22 selected publications (A. Calcante et al., 2014; M. Titler et al., 2015; K.M. Lobeck-Luchterhand et al., 2015; M. Saint-Dizier et al., 2018 and M. V. Rørvang et al., 2018, (Figure 2)) was conducted. Basically, the same criteria were applied for the Pubmed search, but additionally, publications that had already been shortlisted for the Pubmed search (*n* = 29) were excluded. Publications with a title that was off-topic (*n* = 215), published before 1998 (*n* = 15), published in a journal with an IF ≤ 1 (*n* = 1) and not peer-reviewed (*n* = 1) were excluded. In addition, corresponding authors who were cited multiple times in the reference list of the 5 selected publications (*n* = 36) and corresponding authors who are members of our working group (*n* = 3) were excluded. This supplementary search resulted in 19 additional corresponding authors that were included in the final survey.

In total, 80 corresponding authors were invited to participate as subject matter experts (SMEs) in the field of calving management in the survey.

For the non-SMEs, 24 participants in a workshop on bovine perinatology organized by the Ontario Association of Bovine Practitioners (OABP) delivered by J.F. Mee and held in Guelph, Canada on 2nd May 2019 were surveyed as a convenience sample of large animal practitioners.

### 2.3. Administration of the Questionnaire

The same questionnaire was administered to the two veterinary groups; corresponding authors of peer-review papers on calving management [using the online survey software Unipark (https://www.unipark.com/), (online survey) (accessed on 10 July–26 September 2018)] and veterinary participants in a workshop on bovine perinatology [using a paper copy of the questionnaire (workshop survey)].

For the online survey, a cover letter outlining the objective of the research and assuring the participants that all registered data would remain anonymous along with a hyperlink to the questionnaire was sent via personal email addresses to the 80 corresponding authors. The survey was online from 10th July until 26th September 2018 (79 days). The average duration to survey completion time was 20 min. Participation in the survey was voluntary, but participants of the online survey were reminded three times via email. The first reminder was sent out after one week, the second after another week, and the last reminder with the hyperlink for participation after 3 weeks. For the workshop survey, the purpose of the questionnaire was explained to the participants of a workshop on bovine perinatology then it was distributed and collected before the workshop started.

### 2.4. Data Analysis

Data were entered into Microsoft Excel 2004 (Microsoft Corporation, Redmond, WA USA) and analyzed with descriptive statistics. The Likert scales were analyzed using the Mann–Whitney *U*-test to consider the strength of agreement and disagreement. Difference between responses of SME and non-SMEs were calculated using Fisher’s exact test. The significance level was set at *p* < 0.05. Analyses were conducted using IBM SPSS Statistics for Windows (V. 26.0, IBM Deutschland GmbH, Ehningen, Germany).

In the following, only content with a statistical difference in the responses of SME and non-SMEs is mentioned. The responses that did not differ are not mentioned separately.

## 3. Results

### 3.1. Response Rates

Of the 104 participants (80 SMEs and 24 non-SMEs) invited to fill in the questionnaire, 54 questionnaires of the online survey and 24 of the workshop survey were returned, with a response rate of 67.5% and 100%, respectively.

Of the 54 respondents in the online survey, 14 (25.9%) viewed the first page with the introduction of the survey; another 17 (31.5%) canceled the questionnaire early in the process, thus 23 SMEs completed all of the questions. While all workshop participants (non-SMEs = 24) finished the questionnaire, nine did not answer every question. In total, 38 out of 104 (36.5%) complete questionnaires were returned (28.8%—23/80 of the online survey; 62.5%—15/24 of the workshop survey). Incomplete questionnaires were included in the analysis; the data were adjusted to the respective number of participants per question.

### 3.2. General Information about Participants (Question 1–4)

Most participants were from North America (59.4%—15/40 SMEs; 23/24 non-SMEs), followed by Europe (37.5%—24/40 SMEs; 0/24 non-SMEs) and Asia (23.1%—1/40 SMEs; 1/24 non-SMEs). Overall, almost twice as many men as women participated in the survey (65.6% men—23/40 SMEs; 19/24 non-SMEs; 34.4% women—17/40 SMEs; 5/24 non-SMEs). The average number of scientific peer-reviewed publications ranged from 58.4 publications among SMEs to 2.5 publications among non-SMEs (Question 3, open question). The average number of publications related to calving management was 10.1 by SMEs and 0.4 by non-SMEs (Question 4, open question).

### 3.3. Definition of The Stages of Parturition (Question 5–7)

Most participants (89.7%—31/35 SMEs; 21/23 non-SMEs) agreed that the distinction between stage I and II of parturition is helpful in managing calving (Question 5, closed question). A total of 122 descriptors for 12 visible signs to determine stage I of parturition were made by 25 SMEs and 21 non-SMEs (Question 6, open question). Visible signs that were most frequently recommended were “Restlessness” (56.5%—15/25 SMEs; 11/21 non-SMEs), “Tail raising” (50.0%—12/25 SMEs; 11/21 non-SMEs), “Vaginal discharge” (28.3%—7/25 SMEs; 6/21 non-SMEs) and “Relaxation of the pelvic ligaments” (26.1%—9/25 SMEs; 3/21 non-SMEs), (Table 1). SMEs and non-SMEs did not significantly differ in their recommendations for visible signs to determine stage I of parturition.

For the description of stage II of parturition (Question 7, open question), 24 SMEs and 21 non-SMEs gave 95 descriptors and listed 12 recommendable signs to observe (Table 1). The most frequently listed signs were “Visibility of fetal parts in the vulva” (73.3%—16/24 SMEs; 17/21 non-SMEs), “Abdominal contractions” (35.6%—8/24 SMEs; 8/21 non-SMEs), or a “Visible amniotic sac” (35.6%—10/24 SMEs; 6/21 non-SMEs). There was no significant difference between SMEs and non-SMEs.

### 3.4. Observation Routine (Question 8–10, 16 and 17)

Recommendation by 24 SMEs and 24 non-SMEs on when [days pre expected calving date, (ECD)] to observe cows for signs of parturition and potential movement to the maternity pen ranged from more than 21 days to one day before ECD (Question 8, open question, Figure 3). Respondents recommended close observation at least 7 or 10 days (33.3%—7/24 SMEs; 9/24 non-SMEs) before ECD. SMEs and non-SMEs did not differ significantly in this regard.

Recommendations on the daily frequency of observations for signs of impending parturition were dependent on the proximity to parturition (Question 9, semi-open question, Figure 4). This question was answered by 24 SMEs and 24 non-SMEs. For cows that are not yet in the maternity pen, most respondents recommended observation intervals of twice a day (35.4%—8/24 SMEs; 9/24 non-SMEs) and every 6 h (31.3%—6/24 SMEs; 9/24 non-SMEs). Seven participants (14.6%—6/24 SMEs; 1/24 non-SMEs) commented that the observation interval strongly depends on the individual cow and its previous calving performance. The more SIP (signs of imminent parturition) observed the more frequent the number of recommended. As soon as the cow is in the maternity pen (Question 16, semi-open question), the recommended observation intervals were 6 h (45.7%—14/23 SMEs; 7/23 non-SMEs) and 2 h (26.1%—4/23 SMEs; 8/23 non-SMEs). Two participants (4.3%—2/23 SMEs; 0/23 non-SMEs) chose none of the options and stated that the observation interval depends on the individually observable signs of parturition. Answers of SMEs and non-SMEs did not differ significantly.

A majority of study participants recommended moving cows to a maternity pen (81.3%—20/24 SMEs; 19/24 non-SMEs); the other 9 participants (18.8%—4/24 SMEs; 5/24 non-SMEs) disagreed with this practice (Question 10, closed question). The recommendations of SMEs and non-SMEs did not differ significantly.

### 3.5. Signs of Parturition (Question 11, 12, 14, and 15)

Twenty-two SMEs and 21 non-SMEs named 21 important signs of parturition (Figure 5). “Abdominal contractions” (44.2%—19/22 SMEs; 9/21 non-SMEs) “Vaginal discharge” (32.6%—4/22 SMEs; 10/21 non-SMEs) and the “Visibility of fetal parts” (30.2%—8/22 SMEs; 5/21 non-SMEs) were listed most frequently. “Enlargement of the udder” has been listed less frequently, but significantly more frequent by non-SMEs (*p* = 0.04).

In question 11 we offered a choice of ten signs of parturition to determine the right time to move a cow to the maternity pen. The most common responses were “Relaxation of the pelvic ligaments”, (50.0%—14/24 SMEs; 8/20 non-SMEs) and “Behavioral changes” (34.1%—8/24 SMEs; 7/20 non-SMEs) (Figure 6). Thirteen participants, but significantly more non-SMEs (10/20) than SMEs (3/24) (*p* = 0.009) stated “Gestation length” was a recommendable parameter. Some of those recommended 3 weeks (*n* = 5) or 1 week (*n* = 3), others 2 weeks (*n* = 1) or three days (*n* = 1) before ECD. All 39 respondents estimated the predictive value of their recommended SIPs in the upper range (i.e., 41–60%, 61–80% and 81–100%), but only 10.3% of respondents (2/24 SMEs; 2/15 non-SMEs) considered the predictive value of their selection to be very certain (i.e., 81–100%). Most respondents (59.0%—13/24 SMEs; 10/15 non-SMEs) estimated the predictive value of their recommendation at 61–80%. There was no significant difference between SMEs and non-SMEs.

On a 5-point Likert scale, respondents fully agreed that “Lying lateral with abdominal contractions” (83.0%—18/23 SMEs; 21/24 non-SMEs), “Tail raising” (51.1%—16/23 SMEs; 8/24 non-SMEs; *p* = 0.02), and “Vaginal discharge with bloody traces” (48.9%—11/23 SMEs; 12/24 non-SMEs) were very important to check on a regular basis as signs of parturition. “Tripping” was found to be very important by SMEs only (8.5%—4/23 SMEs; 0/24 non-SMEs; *p* = 0.05) (Question 15, closed question, Figure 7).

### 3.6. Influences of Timing of Moving Cows to The Maternity Pen (Question 13, 17, and 18)

When asked to rate on a 5-point Likert scale the importance of influencing factors on the right time of moving a cow to the maternity pen (Question 13, closed question, Figure 8), respondents fully agreed that the expertise of the personnel (62.2%—13/24 SMEs; 15/21 non-SMEs) and the frequency of observation (53.3%—11/24 SMEs; 13/21 non-SMEs) were very important. Sixteen participants (35.6%—9/24 SMEs; 7/21 non-SMEs) fully agreed that the timing of moving cows to the maternity pen relative to the stage of parturition has an influence on the calving process. Most participants fully agreed (24.4%—6/24 SMEs; 5/21 non-SMEs) or agreed (40.0%—7/24 SMEs; 11/21 non-SMEs) that technical devices influence the right time of movement. However, just over half of the respondents (55.3%—15/23 SMEs; 11/23 non-SMEs) recommended devices for calving detection; whereas the others did not. (45.7%—8/23 SMEs; 13/23 non-SMEs). Recommended devices were vaginally inserted temperature or photo loggers for calving detection (30.8%—2/15 SMEs; 6/11 non-SMEs; *p* = 0.04), tail-mounted accelerometers (23.1%—4/15 SMEs; 2/11 non-SMEs) or cameras in the maternity pen (19.2%—2/15 SMEs; 3/11 non-SMEs).

The notion of the influence of cow movement on the vitality of calves is supported by 15 participants (33.3%—7/24 SMEs; 8/21 non-SMEs) who fully agreed and 16 (35.6%—7/24 SMEs; 9/21 non-SMEs) who agreed. The influence of cow movement on the duration of parturition was supported by 27/45 participants, who fully agreed (26.7%) and agreed (33.3%), whereas 8 (17.8%) voted neutral and 4 (8.9%) disagreed, and 2 (4.4%) fully disagreed. Most participants agreed (22.2% fully agreed and 42,2% agreed) that the timing of moving cows to the maternity pen also affects the ease of calving. Answers of SMEs and non-SMEs on influencing factors did not differ significantly.

When asked whether participants would recommend a vaginal examination to rule out dystocia and when they would recommend such an examination (Question 18, semi-open question, Figure 9), the majority (57.4%—11/23 SMEs; 16/24 non-SMEs) recommended that a vaginal examination be performed after the appearance of Amnion Sack (AS) or feet outside the vulva, and of these, 15 participants (55.6%—6/11 SMEs; 9/16 non-SMEs) recommended doing so after 1 h. Four (14.8%—3/11 SMEs; 1/16 non-SMEs) recommended an examination after 2h and 8 (29.6%—2/11 SMEs; 6/16 non-SMEs) after 0.5 h or immediately.

Besides after the appearance of the amniotic sac (AS) or feet of the calf outside the vulva; bloody vaginal discharge (55.3%—9/23 SMEs; 17/24 non-SMEs; *p* = 0.04) or once abdominal contractions (51.1%—8/23 SMEs; 16/24 non-SMEs; *p* = 0.04) have commenced were indices for vaginal examination. Bloody vaginal discharge was considered an immediate indicator for a vaginal examination (37.0%—2/19 SMEs; 8/17 non-SMEs; *p* = 0.02) but others recommended waiting up to 24 h (2.8%—0/19 SMEs; 1/17 non-SMEs). Regarding the interval from observation of abdominal contractions to vaginal examination, the time period with the highest agreement (29.2%—3/8 SMEs; 4/16 non-SMEs) was 1 h, with variation from 30 min to 24 h. Five participants (10.6%—4/23 SMEs; 1/24 non-SMEs) recommended not to perform a vaginal examination at all, only if no calving progress is apparent, bleeding or placenta prolapse.

## 4. Discussion

The overall response rate was high. But it needs to be disaggregated into the online and the workshop survey, complete and incomplete questionnaires, and question type. As expected, the response rate (of complete questionnaires) was higher in the workshop (62.5%) than in the online survey (28.8%). However, the login rate (i.e., the number of participants that started the questionnaire) in the online survey (67.5%) is comparable with other online surveys (e.g., 85% in [30]; 67.3% in [31]; 58% in [32]); the number of participants in the workshop that started the questionnaire was 100%. With online surveys, subscriber loss can occur because individuals cannot distinguish between a legitimate survey and a spam message, even if the emails come from a trusted organization [33]. Reminders to participate and the use of QR (Quick Response) codes were found to significantly increase response rates [34]. In our study, reminders indeed significantly raised the number of participants from the initial 14 (of which only 7 completed the questionnaire) up to 54 (of which only 23 completed the questionnaire).

The distinction between stages I and II of parturition has been determined to be helpful in calving management and is found to be especially important for defining the right time to move cows to the maternity pen. The first stage of parturition is defined in veterinary textbooks [35,36], (i.e., cervix starts opening, myometrial contractions start and the fetus adopts the final position within the birth canal). The duration of this stage is highly variable [37] and has been described from as little as 2 h [38] to 24 h [22] or even has been reported to last up to days [35]. Survey participants recommended for determining this stage, observation of “Restlessness” and “Tail raising”, also “Relaxation of the pelvic ligaments” and “Vaginal discharge”. This is in agreement with Wehrend et al. [39] who recorded “Restlessness” as the most frequently observed behavioral change when parturition was imminent. Berglund et al. [4] included “Restlessness” in their definition of stage I of parturition (i.e., the interval from restlessness until the allantochorion appears). The same time of stage 1 is very flexible, the time when “Restlessness” is observed before parturition varies as well. Huzzey et al. [40] reported a rise in activity one day before calving, which can easily be triggered by other causes such as stress from a different environment [41]. Other authors found “Restlessness” on the day of calving [42] or 12 h [43], 6 h [44,45], 140 min [46], or 120 min before calving. “Restlessness” might be caused by discomfort from labor [47,48] and is expressed by increased standing/lying transitions and increased walking [40,43]. Study participants listed signs, such as “Frequent lying/standing transitions”, “Isolation seeking behavior” and “Behavioral changes” separately, all these terms have been comprised under the term “Restlessness” [3,39]. “Tail raising” and “Relaxation of the pelvic ligaments” were also on the list of signs to identify stage I of parturition of the study participants. “Tail raising” [39,44], “Relaxation of the pelvic ligaments” [3,4,39] are described as visible signs of stage I of parturition. For “Tail raising” time of appearance is reported for 9 h before the onset of stage II, [49] or 6 h before parturition [43,45,48]. The measurement of the relaxation of the pelvic ligaments has been shown to be a useful and accurate tool in predicting parturition within 24 h [23] or no parturition within 12 h [50]. Clear “Vaginal discharge” alone was not considered to be a suitable predictor [49]. Other signs, like “Vulva edematization” [39] or “Olfactory ground checks” [3] were not mentioned by participants.

Signs to describe stage II of parturition named by study participants are consistent with signs reported in the literature [3,18,20,35,51,52]. “Visibility of fetal parts in the vulva” and “Lying lateral with abdominal contractions” were named most frequently as very important signs to check on a regular basis. Abdominal contractions can be increasingly observed about 3 h before parturition [49], so they are seen in stage 1 and stage 2 of parturition.

Recommendations on when best to first begin to observe cows for early signs of imminent parturition and potential movement to the maternity pen are rarely found in the literature. This may be the reason for the large range of suggestions given in our survey. These ranged from one to more than 21 d before ECD. Cook et al. [53] suggested observations of cows in the close-up pen even 14 to 21 d before ECD. The ECD is not a precise metric to move cows into the maternity pen, as the duration of pregnancy varies between 279.4 ± 5.7 [54]. Gestation length can be influenced by the sex of the calf [55], cattle breed [56], twinning [57], and parity of the cow [58]. If moving cows according to ECD, some cows will spend too long in the maternity pen with negative consequences for environmental hygiene [20]. Nevertheless, one SME and 5 of the non-SMEs recommended “gestation length” as the sole parameter to determine the time of moving cows. Possibly, SMEs were more likely to have read the literature providing information on the diverse factors influencing the duration of gestation length [54,58,59] and non-SMEs have experienced in daily routine, that days after insemination works well in practice.

Previous studies report diverse protocols of cow movements before parturition. Cows were moved when parturition was considered imminent, either with no time specification [60,61] or the timing of movement was calculated retrospectively, e.g., within 1–4 h prepartum [62], 4 h before calving [63], or 48 to 72 h [64] up to 7 to 5 d before calving [65].

Respondents prioritized “Relaxation of the pelvic ligaments” and “Behavioral changes” as key criteria when to move cows to the maternity pen. “Relaxation of pelvic ligament” has been described to start as early as 15 d before parturition up to only 7 h ante partum [4]. Measuring the increment in ligament relaxation has a high accuracy (93.9%) in predicting calving within 24 h with and can be easily applied in field conditions [23]. “Behavioral changes” though can be difficult to define. All behaviors such as “olfactory ground checks, nest-building behavior, vocalization, discharge of feces and urine, restlessness, tripping, turning the head towards the abdomen and tail raising” under the term “Behavioral change” [3,39]. The cow’s behavior changes when parturition is approaching [12]; sometimes physiologically due to pain [42,48] and sometimes due to calving difficulties [43]. Sensors measuring behavioral changes for calving prediction, i.e., activity [41] or tail raising [66] are commercially available.

When assessing the predictive value of a given SIP for determining the right time for cow movement, almost all participants chose the upper range between 41 and 100%, but only a very small proportion of participants (10.3%—2/24 SMEs; 2/15 non-SMEs) considered the predictive value of their selection to be very certain (i.e., 80–100%). However, this 10.3% of participants all selected different signs which they named to be very predictive. This suggests over-confidence in experts’ ante judgment, failure to recognize objective ignorance, and perhaps lack of experts post evaluation. The favorite signs rated as important for imminent parturition were “Abdominal contractions” and “Vaginal discharge”. Overall, signs selected were consistent with the literature [3,18,20,35,49,51,52].

Respondents recommended increased frequency of daily observations once the cows were in the maternity pen, most commonly every 2 or 6 h. These recommendations are similar to those published by Mee [37] who suggested observing cows in the first stage of parturition approximately every 3 to 6 h to detect the onset of the second stage or possible calving difficulties. After the onset of the second stage, an observation interval of every 30 min or continuous observation was recommended [37]. However, there are several protocols in scientific studies recommending an observation interval of every two [67] or every hour [3,41,49,65]. It has been shown that poor surveillance during calving leads to a significant increase in stillbirth frequency [68], but a constant presence of an observer can also lead to prolonged calving and dystocia [10]. A survey with farmers in Ireland revealed that only 33% of participants observed their prepartum cows at least every 6 h during the day or night and 24% reported not observing cows at all during the night [30]. Similar findings of poor nocturnal surveillance were reported in studies conducted in Canada [69,70] and in Brazil [71]. Since about half of calvings occur at night, these observation intervals may be inappropriate [72].

The benefits of technical devices for detecting calving and deciding when to move cows were not unanimously accepted by study participants. Various devices for calving surveillance have been developed and evaluated [41,63,73,74]. Study participants most frequently recommended vaginally inserted temperature loggers for calving detection. However, experiences and results of publications were diverse and such devices were not recommended by any author as a sole tool for calving prediction [6,63,73,75,76]. Tail-fixed accelerometers were also recommended by study participants, although there is little published evidence to support their benefits. Studies [66,77] showed poor sensitivity of such a device and welfare issues retaining the sensor to the tail. Some respondents also recommended the use of cameras to monitor calving. Continuous calving monitoring with video cameras has been used in several (research) studies [43,45,48]. However, in practice ‘calving cameras’ are not widely used on commercial dairy farms internationally [30,69,70] and despite the various technological developments, visual observation of cow behavior is the most commonly used approach [6].

When asked about the importance of factors influencing the timing of cow movement to the maternity pen, study participants’ assessments regarding the expertise of the personnel are consistent with the literature where comprehensive training on calving management practices has been identified as a top priority in order to reduce the incidence of calving difficulties [2,3,78]. The influence of the expertise of the personnel, like the influence of observation frequency, was fully agreed or agreed with by 84.4% of the study participants. Obviously the more often an animal is observed, the more likely it is that SIPs will be detected. A high proportion of study participants (80.0%) fully agreed or agreed on the influence of cow movement relative to the stage of parturition. Studies have been shown that there is a sensitive period near the end of stage I of parturition where moving a cow can disrupt calving progress [18,19]. Thus, the authors recommended moving cows during stage II of parturition [19,21] or having the facilities to move cows 2–3 wk earlier [22]. The timing of cow movement is predominantly seen to have an impact on the vitality of calves; 68.9% of study participants fully agreed or agreed. Research showed that the risk for prolonged second stage parturition increased for cows moved late in first stage parturition, resulting in a 2.5-fold increase in stillbirths [19,79]. In addition, studies showed that the duration of parturition significantly influenced the degree of calf vitality [80] and more precisely a study showed that 71% of calves had low vitality when stage II parturition lasted longer than 2 h [81]. 

Most participants agreed (26.66% fully agreed and 33.33% agreed) that the timing of moving cows to the maternity pen has an influence on the duration of parturition. Stress caused by moving the cow to the maternity pen during the first stage of parturition can stop or delay the calving process [22] and cause dystocia [37]. Undisturbed calving is essential because the duration of the second stage of parturition determines the course of calving [79]. Fourteen participants (31.1%) did not consider that, possibly due to the diversity of opinion in the literature [60,61,62,63,64,65] or based on their personal experience.

Most participants agreed (22.2% fully agreed and 42.2% agreed) that the timing of moving cows to the maternity pen also affects the ease of calving. Other studies reported that moving primiparous cows to the maternity pen in stage two of parturition resulted in lower average calving ease scores compared to stage one [21]. 

A vaginal examination can help to rule out dystocia. The majority (89.4%) of study participants recommended such an examination and only five (10.6%) did not recommend examining the cow or only in cases of no calving progress or if blood or the placenta came out. In some studies, cows were given a vaginal examination if no progress of parturition was observed two hours after AS burst [39] or feet were visible [82]. A Canadian study found that 61.3% of farmers do a vaginal examination if the AS already burst but calving is not progressing [69]. In North America, 94.6% of dairy farmers surveyed examined or assisted heifers and cows within 3 h of AS appearance, and 48.4% of those farmers would intervene within 1 h [52]. A high percentage of respondents (77.8%) in the present study recommended an examination within 1 h after observing the AS or feet outside the vulva. Bloody vaginal discharge was considered an immediate indicator for a vaginal examination for 37% of respondents, but others recommended waiting up to 24 h after seeing bloody vaginal discharge. Some of this variation may be due to the imprecise description of bloody vaginal discharge in the question, i.e., neither the volume nor the nature of the discharge was specified.

## 5. Conclusions

The objective of this survey was to elicit current recommendations on calving management from both SMEs and non-SMEs in order to provide farmers with best practice advice for managing the periparturient cow. There was good consensus between SMEs and non-SMEs apart from using ECD as a good sign for moving cows into maternity pens. It seems that some information found in the literature lacked high clinical utility due to the imprecision in either their definition, (e.g., restlessness), measurement (e.g., pelvic ligament relaxation), or timing (e.g., when to move the cow to the calving unit) or their poor evidential base (e.g., calving monitoring devices). 

Thus, it can be concluded that it makes sense to differentiate between stages 1 and 2 of parturition. There is a broad agreement on how to identify imminent signs of parturition, but skilled and motivated personnel is needed to recognize those. However, more research is warranted to determine the best time for moving cows to the maternity pen. To rule out dystocia a vaginal examination of the calving cow was recommended. Technical devices are recommended to identify the onset of calving, without a clear recommendation of the best technique. 

## Figures and Tables

**Figure 1 animals-11-03129-f001:**
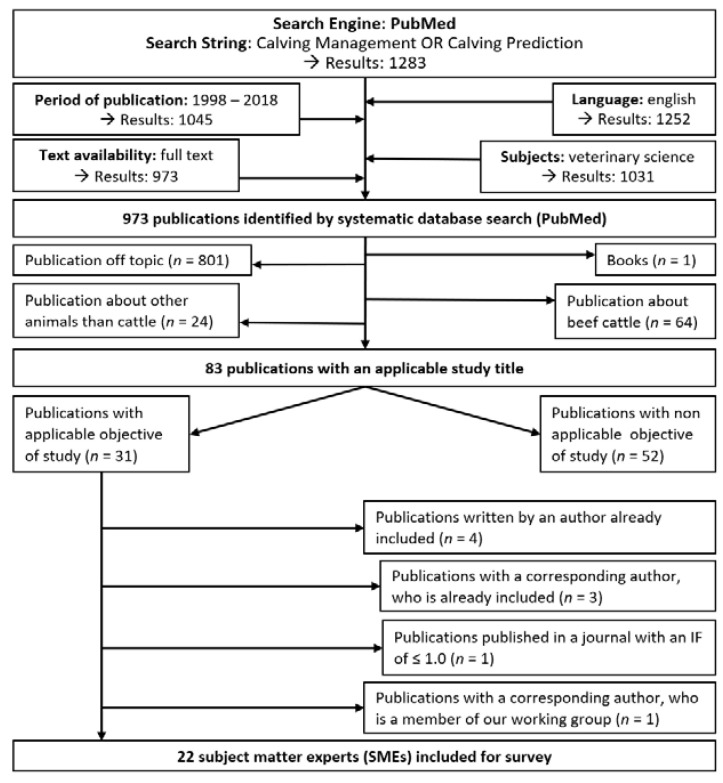
Inclusion and exclusion criteria to identify published subject matter experts (SMEs) in calving management by interrogating the search engine PubMed.

**Figure 2 animals-11-03129-f002:**
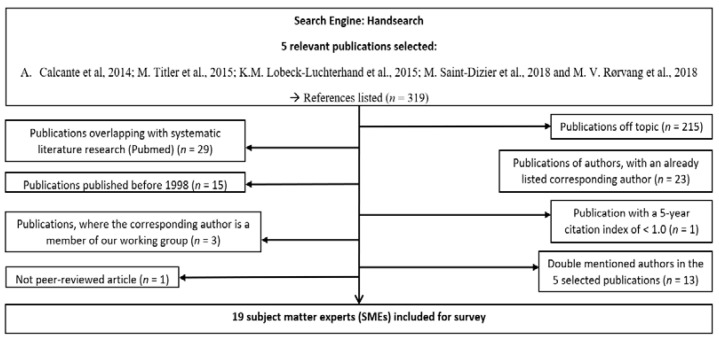
Identifying additional published subject matter experts (SMEs) in calving management by applying inclusion and exclusion criteria on publications listed in the reference lists of 5 relevant publications.

**Figure 3 animals-11-03129-f003:**
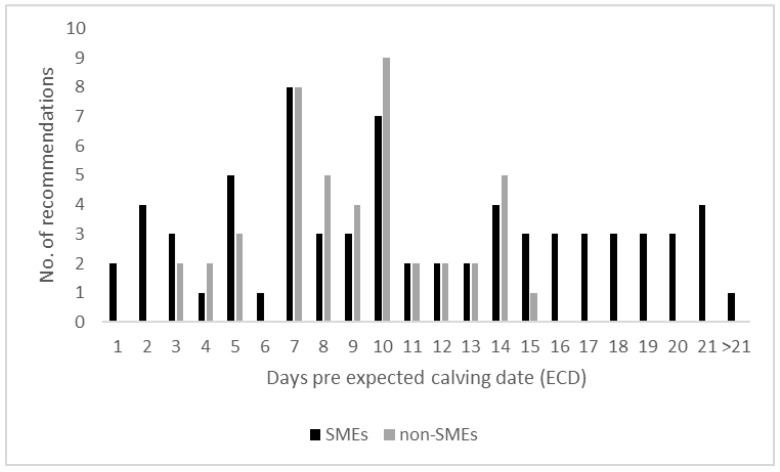
Recommendations for closer observation for signs of parturition of prepartum cows in the dry cow accommodation before movement to the maternity pen (48 respondents—SMEs = 24; non-SMEs = 24).

**Figure 4 animals-11-03129-f004:**
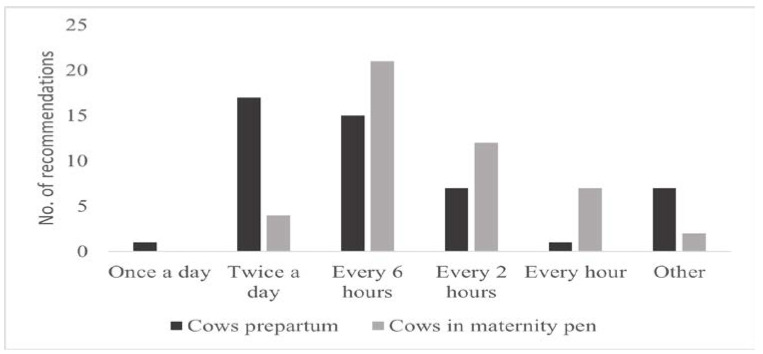
Recommendations on frequency of observations per day for prepartum cows before movement to the maternity pen (48 respondents—SMEs = 24; non-SMEs = 24) and for cows that are in the maternity pen (46 respondents—SMEs = 23; non-SMEs = 23).

**Figure 5 animals-11-03129-f005:**
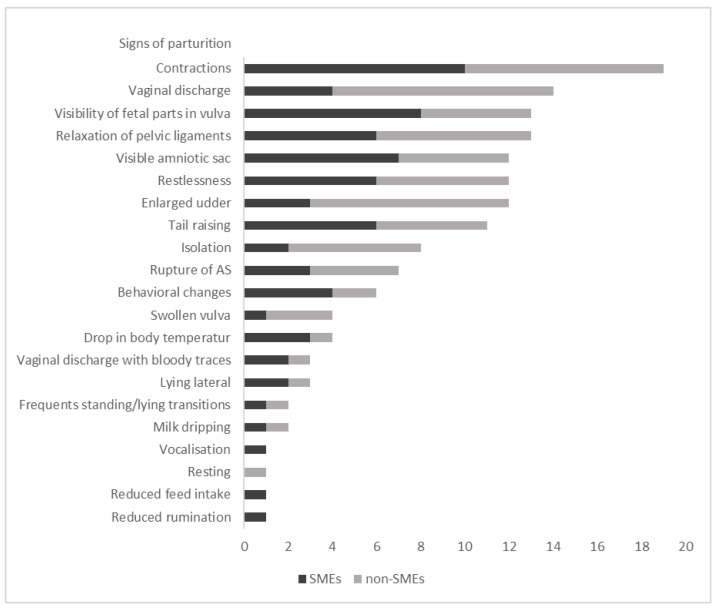
Important signs of parturition (*n* = 21) listed by SMEs (*n* = 22) in an online survey and non-SMEs (*n* = 21) in a workshop survey.

**Figure 6 animals-11-03129-f006:**
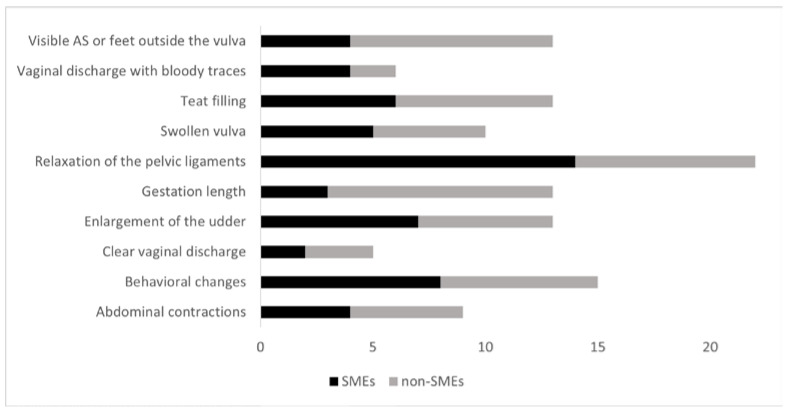
Number of recommendations (alphabetized) for visible signs of parturition to determine the right time for moving a cow to the maternity pen (44 respondents—SMEs = 24; non-SMEs = 20).

**Figure 7 animals-11-03129-f007:**
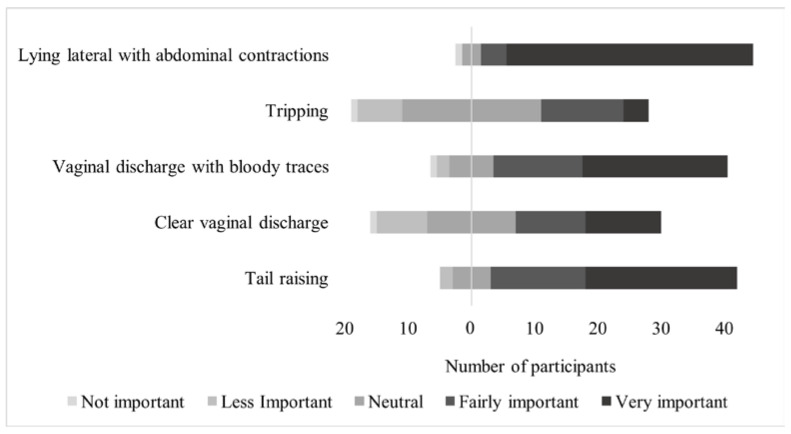
Agreement (5-point Likert scale) with the importance of 5 signs of parturition (47 respondents—SMEs = 23; non-SMEs = 24).

**Figure 8 animals-11-03129-f008:**
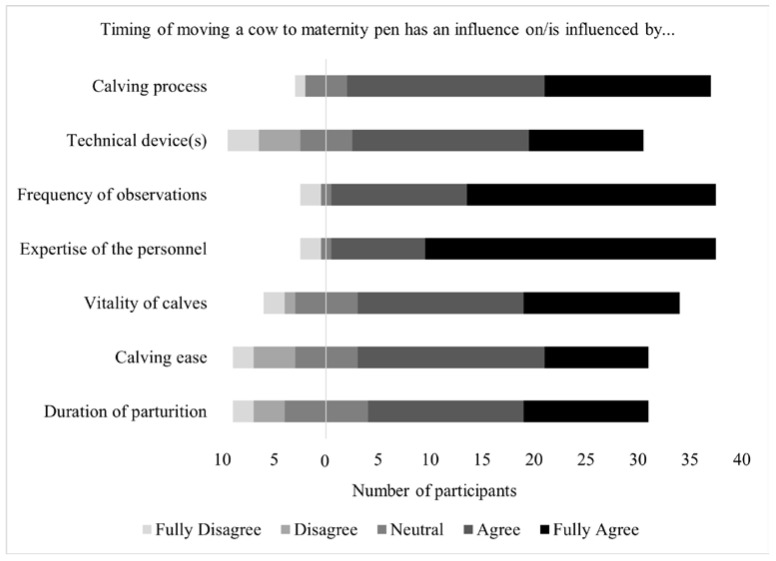
Agreement (5-point Likert scale) with influences on the right time to move cows to the maternity pen (45 respondents—SMEs = 24; non-SMEs = 21).

**Figure 9 animals-11-03129-f009:**
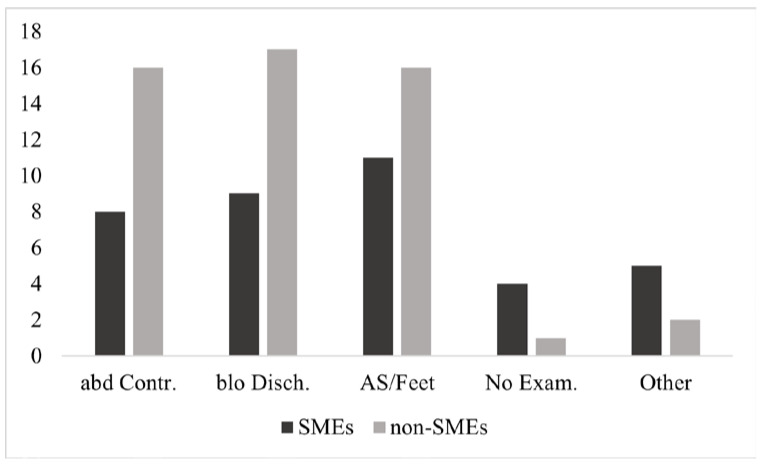
Recommendations on the criteria* used to decide whether a vaginal examination should be performed during calving (47 respondents—SMEs = 23; non-SMEs = 24). *abdominal contractions (abd Contr.), vaginal discharge with bloody traces (blo Disch.), amniotic sac or feet of the calf outside the vulva (AS/Feet), not performing a vaginal examination (No Exam.) or different recommendations (Other).

**Table 1 animals-11-03129-t001:** Number of recommendations for visible signs (alphabetized) to determine stage I and stage II of parturition (Question 6 and 7) from subject matter experts in an online survey and non-subject matter experts in a workshop survey.

Visible Signs of Parturition	Number of Recommendations For			
	Stage I of Parturition		Stage II of Parturition	
	SMEs ^1^ (*n* = 25)	Non-SMEs ^2^ (*n* = 21)	SMEs (*n* = 24)	Non-SMEs (*n* = 21)
Abdominal contractions	5	3	8	8
Behavioral changes	2	1	-	-
Enlargement of the udder	4	1	-	-
Frequent lying/standing transitions	2	0	-	-
Isolation seeking behavior	4	3	0	1
Lateral recumbancy	-	-	2	4
Milk dripping	3	0	-	-
Reduced feed intake	2	3	-	-
Relaxation of the pelvic ligaments	9	3	1	0
Restlessness	15	11	3	0
Rupture of the amniotic sac	2	3	2	4
Swollen vulva	0	3	-	-
Tail raising	12	11	2	2
Tripping	0	1	-	-
Uncomfortable walk	-	-	1	0
Vaginal discharge	7	6	4	2
Vaginal discharge with bloody traces	-	-	2	0
Visibility of foetal parts in vulva	2	1	16	17
Visible amniotic sac	0	0	10	6
Vocalization	0	1	-	-

^1^ SMEs = subject matter experts. ^2^ non-SMEs = non-subject matter experts.

## Data Availability

Please refer to suggested Data Availability Statements in section “MDPI Research Data Policies” at https://www.mdpi.com/ethics.

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
