# Peer review of "Calving Management: A Questionnaire Survey of Veterinary Subject Matter Experts and Non-Experts"

_animals, 2021, doi:10.3390/ani11113129_

Round 1

Reviewer 1 Report

Interesting topic, and different approach to find solutions.

Very small sample size.

Too much length about filtering in internet.

Reference 26 and 27? Typographical mistake probably

Author Response

First of all, we would like to thank the reviewer for their helpful comments and valuable questions. Please find our specific comments and description of changes below. All changes made to the manuscript have been highlighted in yellow. As a result of these changes, the manuscript improved considerably – thank you!

For a point by point response, please see the attachement.

Best regards,

Carola Fischer- Tenhagen

Reviewer 2 Report

- The aim of the study is debatable, because in social life both groups of the respondents are often not only not isolated from each other, but also know each other (e.g. from publications), usually cooperate with each other and exchange information in accordance with the principle of not neglecting contacts between theoretical and practical veterinarian knowledge. In this context, an important question is: did SMEs and non-SMEs knew their views of each other before participating in the survey? In this sense, in my opinion, the purpose of the work may be problematic. However, I am awaiting clarification from the Authors in this case.
- How can you be sure that in the case of people from the SMEs group, the questionnaires were not completed by other people / random persons, since the survey was conducted anonymously and using an online method?
- Graph 4 is missing from the manuscript therefore difficult to refer to the results in this case.
- The authors note that the opinions of SMEs and non-SMEs were similar to each other. I'm not so sure. For example, Fig. 1 (SMEs response range from 3 to 15 days, non-SMEs response range from 1 to over 21 days); Fig. 5 (significant disparities between SMEs and non-SMEs in the case of vaginal discharge, enlarged udder, isolation, swollen vulva, and other less frequent signs of parturition) and many signs in Fig. 6.
- Why did Figures 7 and 8 not distinguish SMEs and non-SMEs separately? It is important in the context of the aim of the study.
- Excellent discussion - congratulation.

Author Response

(The authors gave the same response as above.)

Reviewer 3 Report

The manuscript covers a survey study carried with the objective to evaluate differences between experts and non-experts regarding the management of cows around peripartum. The manuscript is well written and well structured.

General comments:

The term prediction is used throughout the paper to characterise indicators or parameters that can be used to correctly detect or identify the onset of calving. I do not believe “prediction” would be the correct term in most of the occasions. This term mostly refers to a prediction carried out by a model. I would suggest changing it for wither detection or identification in the manuscript.

Specific comments:

Line 14: add space on “the two”

Line 19: not prediction. Replace it for detection

Line 52: Misplaced parenthesis

Line 52-53: Replace predicted for expected

Line 68: Misplaced parenthesis

Line 119: Google scholar is not a database. The search you carried out on that website, if repeated in another device, will give different results. Therefore, it is not reproducible.

Figure 1. Why the diagram only shows the criteria for PubMed results and not for Google Scholar?

Line 143: Percentage of experts who responded the survey should be in the Results section only.

Line 172 to 174: I believe you meant statistically significant difference. This should be fixed. In addition, it is not clear in your results sections what are the ones that differed and the ones that did not differ.

Figure 4. The figure is missing.

Line 342 – 344. Should be merged with the next paragraph

Author Response

(The authors gave the same response as above.)

Round 2

Reviewer 2 Report

Dear Authors,

Thank You for the explanations and text corrections.

Best regards

The reviewer.